# Earlier onset of North Atlantic hurricane season with warming oceans

Ryan E. Truchelut [1✉], Philip J. Klotzbach [2], Erica M. Staehling [1,3], Kimberly M. Wood[4], Daniel J. Halperin[5], Carl J. Schreck III [6] & Eric S. Blake[7]

Numerous Atlantic basin tropical cyclones have recently developed prior to the official start of hurricane season, including several pre-season landfalls in the continental United States. Pre-season and early-season tropical cyclones disproportionately affect populated land-masses, often producing outsized precipitation impacts. Here we show a significant trend towards earlier onset of tropical cyclone activity in the North Atlantic basin, with threshold dates of the first three percentiles of accumulated cyclone energy shifting earlier at a rate exceeding five days decade[−1] since 1979, even correcting for biases in climatology due to increased detection of short-lived storms. Initial threshold dates of continental United States named storm landfalls have trended earlier by two days decade[−1] since 1900. The trend towards additional pre-season and early-season activity is linked to spring thermodynamic conditions becoming more conducive for tropical cyclone formation. Genesis potential index value increases in the western Atlantic basin are primarily driven by warming ocean temperatures.

[1] WeatherTiger, LLC, Tallahassee, FL, USA. [2] Department of Atmospheric Science, Colorado State University, Fort Collins, CO, USA. [3] Office of STEM Teaching Activities, Florida State University, Tallahassee, FL, USA. [4] Department of Geosciences, Mississippi State University, Mississippi State, MS, USA. [5] Department of Applied Aviation Sciences, Embry-Riddle Aeronautical University, Daytona Beach, FL, USA. [6] North Carolina Institute for Climate Studies, Cooperative Institute for Satellite Earth System Studies, North Carolina State University, Asheville, NC, USA. [7] National Hurricane Center, National Oceanic and Atmospheric Administration, Miami, FL, USA. ✉email: ryan@weathertiger.com

The current operational definition of the Atlantic hurricane season (1 June–30 November) was chosen in 1965 to capture the vast majority of tropical cyclone (TC) formations in the North Atlantic basin[1], reflecting the strongly peaked seasonal cycle of Atlantic TC activity[2]. The brief overlap of vertical instability, mid-level relative humidity (RH), and vertical wind shear (VWS) conditions favorable for TC genesis is responsible for this intraseasonal distribution[3], with over 85% of accumulated cyclone energy (ACE; see Eqs. 1 and 2 in Methods) occurring between 1 August and 31 October. However, the concept of a "hurricane season" lacks a precise definition, and over 2012–2020, seven TCs developing before 1 June required tropical storm watches or warnings for the continental United States (CONUS).

Understanding trends in TC activity onset is important, as early-season storms disproportionately impact populated areas, especially through excessive precipitation[4,5]. Over 1979–2020, 41% of North Atlantic TCs developing before 1 August made landfall in the CONUS as tropical storms or hurricanes (i.e., maximum sustained winds > = 34 kt), compared to 23% of all other TCs. While only 6% of basin-wide ACE occurs prior to 1 August, 17% of ACE over or near the CONUS (USACE) occurs prior to 1 August[6]. Prior studies of Atlantic TC season onset yielded mixed results; one suggested initial named storm formation in a portion of the central and eastern Atlantic shifted earlier by ~1 day year$^{-1}$ over 1980–2007[7], and another found no broader trend for 1979–2014[8]. A study investigating projected changes in Atlantic TC season length obtained conflicting results from different models[9].

Technological advances in Atlantic TC detection complicate trend assessment, as the historical record is likely missing 1–3 TCs year$^{-1}$ prior to 1950[10], and short-duration TCs are more common since 2000[11]. To address these biases in climatology, we focus on Atlantic ACE in the era of geostationary satellite coverage, high-quality reanalyses, and the consistent inclusion of subtropical cyclones (1979–2020). We also examine running averages of ACE since the advent of aircraft TC reconnaissance (from ~1950). The CONUS landfall record is reliable from 1900 due to sufficient coastal observation density[12], with no long-term interannual trend[6], so we use USACE to provide an impacts-based perspective on variations in TC risk onset. In all analyses involving ACE and storm count, we performed regressions including as well as excluding short-lived TCs to quantify the potential influence of observation bias.

To account for interannual and decadal variance in Atlantic TC activity arising from a combination of natural internal variability[13] and external greenhouse gas and aerosol forcing[14,15], we use quantile regression to investigate trends in the intraseasonal distribution of ACE and USACE. Quantile regressions estimate how the expected date of reaching each ACE and USACE percentile threshold has changed while controlling for variability in overall TC activity and limiting the influence of outliers. This method has been used to examine changes in TC formation date[7,8], maximum intensity[16], and intensification rate[17]. ACE and USACE are well-suited for quantile regression because energetic metrics are continuous and more heavily weight longer-lasting, intense, and societally impactful storms.

Using these techniques, here we show a significant trend towards earlier onset of North Atlantic TC activity. Threshold dates of the first three percentiles of ACE shift earlier at a rate exceeding 5 days decade$^{-1}$ since 1979, and by ~2 days decade$^{-1}$ for initial CONUS. named storm landfall threshold dates since 1900. Quantile regression analysis shows that this additional pre-season and early-season activity is linked to spring thermodynamic conditions becoming more conducive for TC formation, as measured by several genesis potential indices, with these changes primarily driven by warming ocean temperatures in the western Atlantic basin. This research supports revising the bounds of the operational Atlantic hurricane season to potentially include portions of May.

## Results

**Season onset trend assessment**. We interrogate potential changes in Atlantic TC season onset with time using several methods. The simplest is to examine the record of initial storm formation and landfall dates. The development of the first Atlantic named storm is trending earlier at a rate of 1.2 days year$^{-1}$ ($p < 0.005$) over 1979–2020 (Supplementary Fig. 1), or 1.4 days year$^{-1}$ ($p < 0.001$) excluding short-lived TCs[11]. The first CONUS landfall has trended earlier at a rate of 0.22 days year$^{-1}$ over 1900–2020 ($p < 0.05$). This trend has reduced the proportions of both TC occurrence and TC impacts within the official Atlantic hurricane season. Over 1971–2020, 96% of CONUS landfalls occurred between 1 June–30 November, compared with 99% during 1921–1970 (Supplementary Fig. 2). In-season named storm formations fell to a low of 96%, and in-season named storm days, ACE, and USACE also declined to near-minima for 1971–2020 means.

Regressions of Atlantic ACE percentile threshold dates against year over 1979–2020 show no significant shift in the distribution of activity over most of the hurricane season (Fig. 1a). For instance, there is no trend in the date Atlantic ACE is expected to reach the 10%, 50%, or 90% completion thresholds. However, the 1st–3rd ACE percentile threshold dates trend earlier at a rate of 0.5–1.0 day year$^{-1}$ ($p < 0.05$ for 2nd–3rd percentiles, $p < 0.01$ for 1st percentile) over 1979–2020. Excluding short-lived TCs (Supplementary Fig. 3), the trend strengthens to ~1 day year$^{-1}$ earlier for the 1st–3rd ACE percentiles ($p < 0.01$). For CONUS landfalls, the 1st–2nd USACE percentile threshold dates over 1900–2020 are trending earlier by 0.2 days year$^{-1}$ ($p < 0.05$). The close agreement between initial percentile threshold trends and the season onset trendlines (Supplementary Fig. 1) suggests the primary importance of initial formation and landfall events on observed shifts in the first few percentile threshold dates.

Although trend detection can be sensitive to timeseries length[8,18], shifts in season onset are robust across variations in initial year of the quantile regression. Changing the start year and fixing the end year at 2020, Atlantic ACE 1st percentile threshold date trends earlier at a rate of 0.15–0.5 days year$^{-1}$ for starting years 1950–1965, and 0.5–1.0 days year$^{-1}$ ($p < 0.05$) for start years 1965–1987 (Fig. 1b). Results are substantially similar when excluding short-lived TCs, with trends of 0.5–1.5 days earlier year$^{-1}$ ($p < 0.05$) for starting years after 1970 (Supplementary Fig. 3). Trends are significant for USACE 1st percentile threshold date for all windows, shifting 0.2–0.75 days year$^{-1}$ earlier ($p < 0.05$) for 1900–1969 start years, increasing to 0.75–2.0 days year$^{-1}$ earlier ($p < 0.01$) for 1970–1990 (Fig. 1c) start years.

**Quantifying environmental changes**. To assess the physical mechanisms underlying these shifts in Atlantic ACE and USACE distribution, we consider the spring synoptic environment in the western Atlantic (WATL) basin (10–36°N, 100–70°W) where pre-season and early-season TC development is most common (Supplementary Fig. 4). TC genesis depends on sea surface temperature (SST), mid-level RH, conditional instability, low-level relative vorticity, VWS, and the Coriolis force[19]; the first three of these are thermodynamic terms, and the latter three are dynamic terms. Genesis potential indices condense physical parameters such as these into single metrics capturing the spatial, intraseasonal, and interannual variability of TC climatology, which is useful for estimating genesis and landfall risks[20]. Here we use the ECMWF Reanalysis v5[21] and Extended Reconstructed

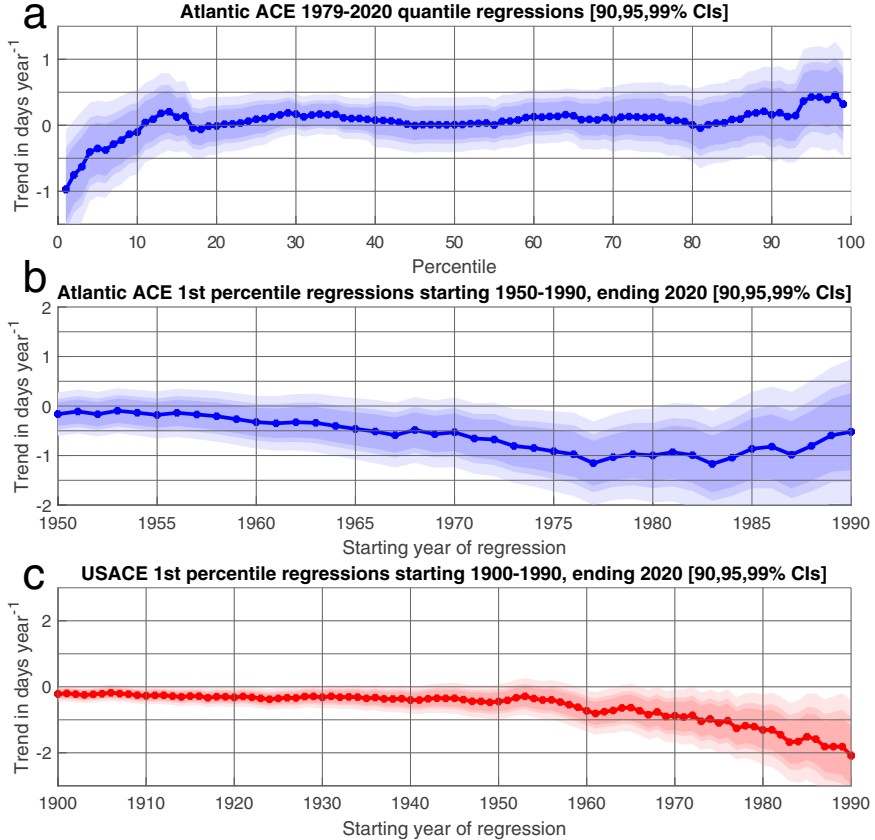

**Fig. 1 Quantile regressions of Accumulated Cyclone Energy (ACE) and ACE over the continental United States (USACE) with year. a** Trend in days year$^{-1}$ for Atlantic ACE 1st–99th percentile threshold dates between 1979–2020. Light, medium, and dark blue shading designate 99%, 95%, and 90% confidence intervals. **b** Trend in days year$^{-1}$ of 1st percentile threshold dates of ACE for starting years 1950–1990 and ending year 2020. Shading as in Fig. 1a. **c** As in Fig. 1b, with USACE and starting years 1900–1990. Light, medium, and dark red shading designate 99%, 95% and 90% confidence intervals (CI).

SST, version 5[22] datasets to examine the relationship between ACE and USACE threshold dates, and Genesis Potential Indices[23,24] (GPIs; see Eqs. 3, 4, and 5 in Methods) and their constituent environmental factors associated with TC formation[3].

Quantile regressions of Atlantic ACE percentile threshold dates against mean WATL spring (April–May) values of GPI[23] are shown in Fig. 2a. Higher spring GPI values are linked to earlier threshold dates for the first ten percentiles of ACE ($p < 0.05$) at a rate of 20–30 days earlier per marginal unit of GPI in the region where season onset TC activity is most frequent. Excluding short-lived TCs causes little change to the relationship between initial ACE percentile threshold dates and spring GPI (Fig. 2c). Repeating this regression for USACE yields similar results (Fig. 2b), with higher spring GPI associated with earlier threshold dates for the first three percentiles of USACE ($p < 0.05$). Substituting a different formulation of GPI[24] does not change the significance of the relationship between spring GPI and 1st–3rd ACE and USACE percentile threshold dates (Supplementary Table 1).

Figure 2d shows a significant trend toward WATL GPI[23] values that are more conducive for early-season cyclogenesis since 1979 ($p < 0.05$). With WATL spring GPI increasing ~0.1 units decade$^{-1}$ and quantile regression coefficients of ~25 days unit$^{-1}$, GPI changes are linked to a shift in initial ACE percentile threshold dates of 10–15 days earlier over 1979–2020. Similarly, a quantile regression coefficient of ~40 days unit$^{-1}$ for WATL spring GPI[23] onto USACE first percentile threshold date is related to a shift earlier of 15–20 days in initial CONUS landfall since 1979

(Supplementary Table 1). For ACE, the corresponding implied shift earlier in onset using GPI[24] is around half of these values due to this index having a smaller positive trend over 1979–2020.

The overall shift to a more favorable environment for early-season TC genesis arises from individual changes in the thermodynamic and dynamic genesis parameters[19] included in the GPIs[23–25]. Following previous attribution studies of TC genesis[3,19,23], we repeat the ACE and USACE quantile regressions for each GPI component, using SST and 200-hPa temperature (200 T) as surface- and outflow-level proxies for potential intensity[26]. These regressions (Fig. 3a–d) show that the relationship between season onset and spring GPI is primarily modulated by the thermodynamic terms: higher WATL spring SSTs and RH are associated with significantly earlier threshold dates for the first three ACE percentiles ($p < 0.01$). Excluding short-lived TCs further strengthens the relationship of season onset with SST and does not alter the significance ($p < 0.05$) of the relationship with RH (Supplementary Fig. 5). Higher SSTs and RH are linked to earlier USACE thresholds for the first four and first eighteen percentiles ($p < 0.05$), respectively (Supplementary Fig. 6). There is no significant relationship between initial ACE or USACE percentile threshold dates and spring WATL 200 T or VWS.

To estimate the marginal change in Atlantic TC season onset attributable to each genesis parameter, we multiply the 1st percentile ACE and USACE threshold date quantile regression coefficients for each genesis parameter (Fig. 3a–d) by the WATL trend of each parameter over 1979–2020 (Fig. 3e–h). As each additional °C of WATL spring SST warming is expected to move

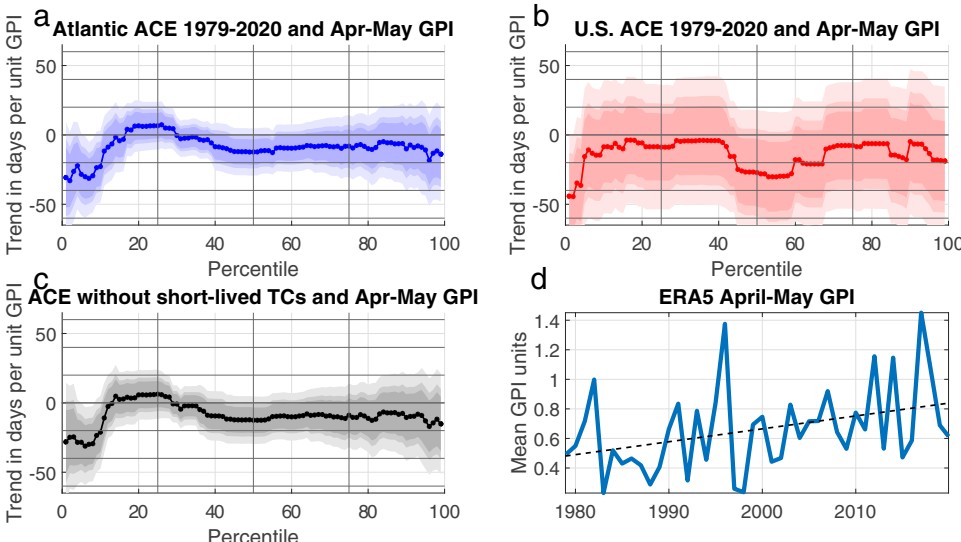

**Fig. 2 Quantile regressions of tropical cyclone (TC) activity with western Atlantic basin spring Genesis Potential Index (GPI) values.** Trends of 1st–99th percentile threshold date between 1979–2020 in days per April–May mean unit of GPI[23] over 10–36°N, 100–70°W for (**a**) Atlantic Accumulated Cyclone Energy (ACE), (**b**) ACE over the continental United States (USACE), and (**c**) Atlantic ACE excluding short-lived TCs. **d** Mean values of April-May GPI from ERA5 reanalysis over 10–36°N, 100–70°W for 1979-2020, with the linear trend displayed with a dashed line. Light, medium, and dark shading indicates the 99%, 95%, and 90% confidence intervals, as in Fig. 1.

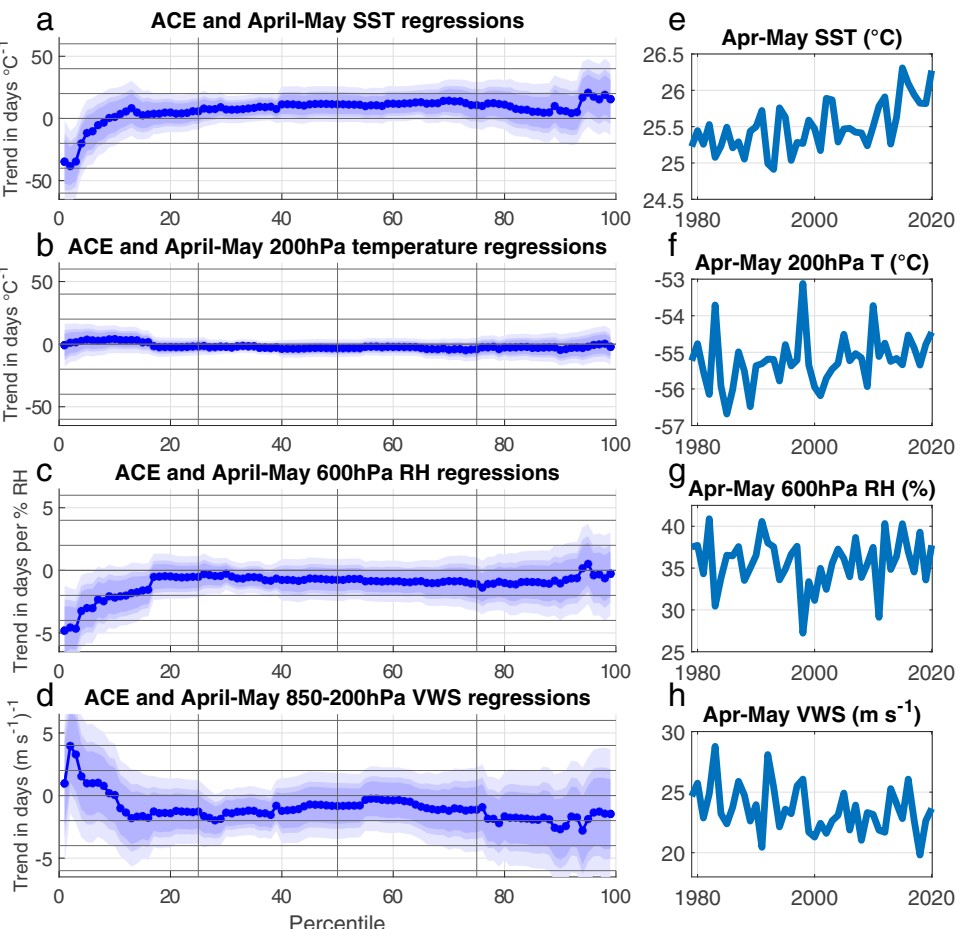

**Fig. 3 Quantile regressions of Accumulated Cyclone Energy (ACE) with genesis parameters.** Trends of Atlantic ACE 1st–99th percentile threshold date between 1979–2020 in days per April–May mean (**a**) sea surface temperatures (SST; °C), (**b**) 200 hPa temperatures (°C), (**c**) 600-hPa relative humidity (RH; %), (**d**) 850-200-hPa vertical wind shear (VWS; m s⁻¹) over 10–36°N, 100–70°W. Mean values of April-May (**e**) ERSSTv5 SST (°C), (**f**) ERA5 200 hPa temperatures (°C), (**g**) ERA5 RH (%), and (**h**) ERA5 VWS (m s⁻¹) over 10–36°N, 100–70°W for 1979–2020. Light, medium, and dark blue shading in (**a**–**d**) indicates the 99%, 95%, and 90% confidence intervals, as in Fig. 1.

occurrence- and impacts-based season onset ~35 days sooner, observed SST changes alone imply a shift around 25 days earlier for the 1st percentile ACE and USACE threshold dates over 1979–2020. Threshold date shifts due to 200 T, RH, and VWS are an order of magnitude lower than for SST and are not significantly different from zero for either ACE or USACE (Supplementary Table 1). The predicted shifts in occurrence- and impacts-based measures of season onset linked to changes in SST broadly match the observed 20-day difference between the median date of the first ACE percentile over 1979–1999 (6 July) and 2000–2020 (16 June), and the 31-day difference in the median 1st percentile USACE threshold dates (22 July and 20 June) for the same periods.

## Discussion

These results show a significant change in spring WATL GPI over 1979–2020. Further interrogating the causes of these GPI changes, our findings support earlier research suggesting that instability and RH modulate a broad slate of occurrence- and impacts-based measures of Atlantic TC season onset[3]. While both spring WATL SST and RH influence initial ACE and USACE threshold dates, only SST shows a significant trend since 1979, with no observed change in RH. Furthermore, warmer detrended April-May mean WATL SSTs are significantly linked to earlier detrended 1st percentile ACE threshold dates after removing short-lived TCs (not shown; $p < 0.05$). This finding provides evidence that the relationship between SSTs and season onset exists independently of recent trends. Thus, increasing upper-ocean temperatures are likely the primary driver of additional pre-season and early-season Atlantic TC activity in recent decades, mostly manifesting as a shift earlier in initial TC formation and CONUS landfall dates.

While a portion of the increased instability driven by higher SSTs (Fig. 3e) is offset by 200 T warming (Fig. 3f), both WATL SST and WATL 200 T are increasing by ~0.15 °C decade$^{-1}$ since 1979. For moist adiabatic lapse rate adjustment to yield constant column stability, rates of upper-tropospheric warming need to be approximately double that of SSTs[27]. Therefore, reanalysis data indicate an increase in mean spring WATL instability over 1979–2020. In the 2010s, the majority of the WATL reached the approximate convective threshold for TC genesis[28,29] in non-baroclinic synoptic environments[30] ~15 days prior to the majority of the WATL reaching that threshold before 1990 (Supplementary Fig. 7). Pre-season TC formation also preferentially occurs in years with greater spatial extent of WATL thermodynamic conditions permissible for genesis in May, likely due to the increased probability a precursor disturbance will enter an environment conducive to further organization.

Quantile regression shows spring WATL SSTs are strongly associated with more pre-season and early-season ACE and USACE, roughly explaining observed changes in season onset over 1979–2020. Spring WATL SST warming trends have accelerated to 0.3 °C decade$^{-1}$ since 1995 (Supplementary Fig. 4), suggesting that the threshold dates for the earliest percentiles of ACE and USACE might continue to shift earlier, irrespective of any changes in annual activity. While these initial percentiles comprise a small proportion of total Atlantic TC activity, early season TCs can have outsized societal impacts that do not neatly map to any quantitative metric; Hurricane Agnes (1972) and Tropical Storm Allison (2001) are two satellite-era examples of extreme TC precipitation events within the opening weeks of hurricane season.

Figure 4 shows the objectively-determined hurricane season start and end dates that most compactly encompass several thresholds of various Atlantic TC activity metrics using 50-year running averages. This quantitative approach to defining season length demonstrates long-term trends earlier in optimized start dates based on initial Atlantic named storm formations, initial CONUS named storm landfalls, and USACE, all of which suggest hurricane season could empirically be defined as beginning prior to 1 June based on 1971–2020 climatology. Additional SST increases may exacerbate the existing mismatch between the effective annual onset of TC risks to populated landmasses and the current operational start of the hurricane season, potentially moving the initial date of Atlantic TC formation earlier by ~0.5–1.0 days year$^{-1}$.

## Methods

**Historical tropical cyclone data.** All historical data for Atlantic tropical cyclones are taken from the HURDAT2 dataset[2] for 1900–2020. Historical CONUS tropical storm watch and warning information is found in the National Hurricane Center Tropical Cyclone Reports for each 1 April to 31 May TC forming over the period 2012–2020 for which CONUS watches or warnings were required[31–37].

ACE is an integrated metric accounting for frequency, intensity and duration of storms. ACE[38] calculations are performed from the six-hourly HURDAT2 intensity data, with the ACE contribution for each six-hourly maximum sustained wind v in knots calculated:

$$ACE = \frac{v^2}{10000} \tag{1}$$

and all entries for which a TC or subtropical cyclone (STC) has v >= 34 knots summed to yield annual totals. Extratropical cyclone data are not considered.

USACE calculations are performed as described in the methodology of ref. 6. Beginning from the HURDAT2 data, we consulted the Atlantic Oceanographic and Meteorological Laboratory hurricane landfall list[39] to identify systems that impacted the CONUS during 1966–1981 when HURDAT2 does not explicitly include landfall points. We then used existing track maps to qualitatively estimate a landfall position and intensity data point to the nearest hour, which we manually added to HURDAT2 dataset for these storms. These position and intensity data were linearly interpolated to one-hour temporal resolution. We use this hourly data set to calculate hourly ACE from maximum sustained winds v in knots by:

$$ACE_h = \frac{v^2}{60000} \tag{2}$$

To yield an annual time series of USACE, we applied a spatial land mask to this hourly ACE metric to exclude all TC positions farther than 0.5° from any part of the CONUS. We selected the 0.5° buffer to correspond to the typical radius of maximum winds of a mature TC and to account for possible small observation errors, particularly early in the record[40]. For the purpose of counts, we consider a "landfall" as the center of circulation of a TC or STC with v >= 34 knots passing within 0.5° of CONUS for any one-hour interpolated timestep. An individual TC or STC may make multiple CONUS landfalls, but only may make a maximum of one landfall in each of the coastal regions shown in Fig. 4a from ref. 6. To find annual values of USACE, we summed this series over the CONUS and temporally over each hour of the year. All timeseries, distributions, and counts in this paper consider the annual cycle of TC activity as beginning at 0000 UTC on 1 March of the given year and terminating at 2359 UTC on 28 or 29 February of the succeeding year. This choice is made to best align with the mean annual minimum of North Atlantic SSTs, and therefore January and February TC activity such as 2016's Hurricane Alex[41] is considered to be part of the previous year's annual cycle of North Atlantic TC activity. January–February TC and STC formations are less than 0.5% of total annual formations over 1950–2020.

*Quantile regression and trend calculations.* Trends in initial Atlantic named storm date were calculated from ordinary least squares regression of the annual series of the number of six-hour periods between 0000 UTC 1 March and the first six-hourly HURDAT2 TC or STC with sustained winds >= 34 knots against year, subsequently converted back into a rate of days year$^{-1}$. This calculation was repeated to exclude any TC or STC with fewer than 8 six-hourly HURDAT2 entries with maximum sustained winds >= 34 knots in order to test the influence of short-lived systems on the overall trend. Trends in initial CONUS impact date were calculated from ordinary least squares regression of the annual series of the number of one-hour periods between 0000 UTC 1 March and the first one-hourly interpolated HURDAT2 TC or STC entry with sustained winds >= 34 knots passing within 0.5° or less of CONUS against year, converted back into a rate of days year$^{-1}$. The significance of all trends was assessed with a Mann–Kendall test[42,43]. Proportions of TC activity occurring within the current official definition of hurricane season are reported as the percentage of all 1 March–28/29 February TC and STC initial formations with sustained winds >= 34 knots, CONUS TC and STC landfalls with sustained winds >= 34 knots, named storm days, ACE, and USACE, occurring between 0000 UTC 1 June and 2359 UTC 30 November. These data are calculated in 50-year trailing average windows to account for multi-decadal variability in Atlantic TC activity[13].

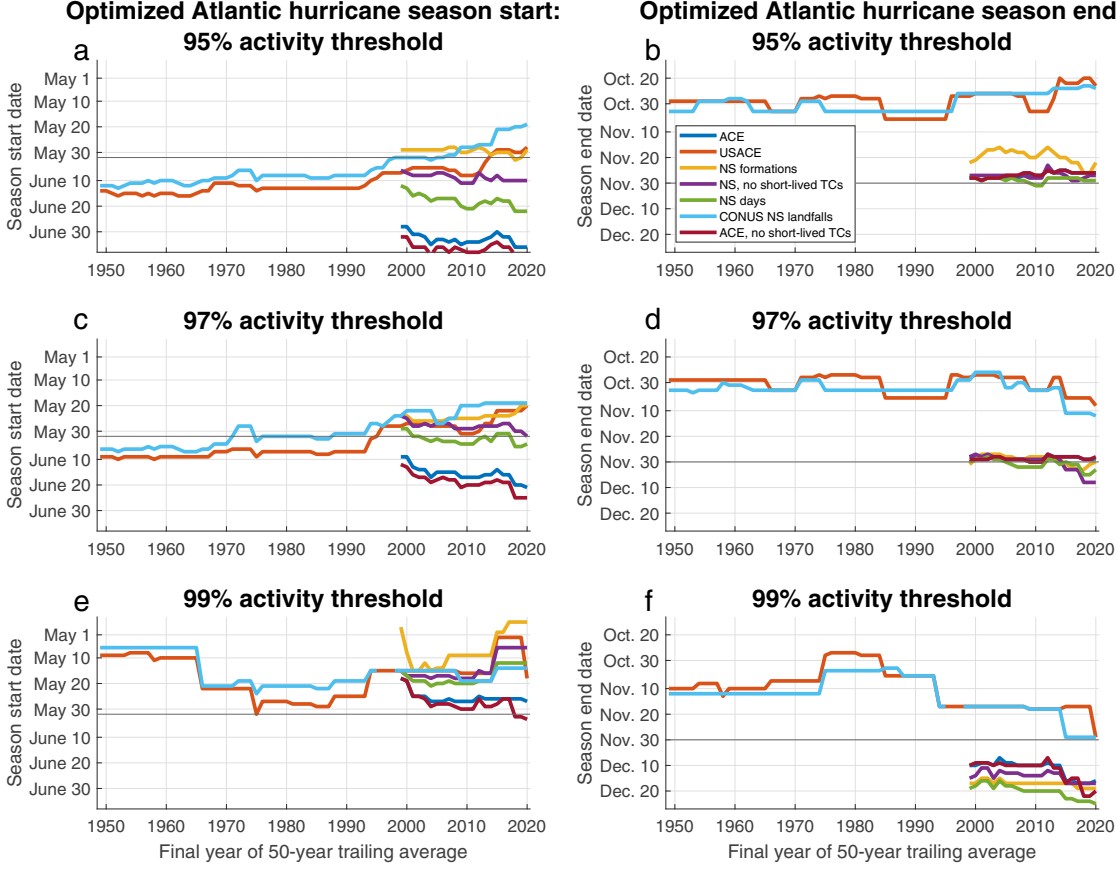

**Fig. 4 Objectively determined Atlantic hurricane season bounds. a, c, e** Starting and (**b, d, f**) ending dates of the most compact timeframe capturing (**a, b**) 95%, (**c, d**) 97%, and (**e, f**) 99% of tropical cyclone (TC) activity, as defined by Atlantic Accumulated Cyclone Energy (ACE), ACE over the continental United States (USACE), named storm (NS) formations, named storm formations excluding short-lived TCs, named storm days, continental United States (CONUS) named storm landfalls, and Atlantic ACE excluding short-lived TCs. Objective calculation of dates are performed using timeseries with 50-year trailing averages of activity and 15-day smoothing of intraseasonal climatology. Trailing averages begin with 1900–1949 for landfalls and USACE, and 1950–1999 for all other metrics.

Trends in percentile threshold dates for ACE (both a climatology in which short-lived TCs are included, and one in which they are excluded per the criteria above) and USACE are found using quantile regression[44]. Quantiles divide the threshold dates at which the specified percentage of total annual ACE and USACE is reached, here ranging from 1% to 99% at intervals of 1%, into equal subsets. Quantile regression applies ordinary least-squares regression to conditional quantiles of the ACE and USACE response variables, which are taken at regular intervals from the cumulative distribution of ACE and USACE over 1979–2020 and 1900–2020, respectively, for regressions performed against year. The sensitivity of the calculated trend to the choice of timeseries length is tested by repeating this quantile regression methodology for initial years of the cumulative distribution functions for ACE of 1950–1990 and USACE of 1900–1990, with a fixed ending year of 2020.

*Quantile regression synoptic environment sensitivity tests.* The dependence of ACE and USACE percentile threshold dates on synoptic environmental factors is tested by performing quantile regressions of these response variables against GPI, SST, 200 T, RH, and VWS. These regressions are performed from the cumulative distribution functions of ACE and USACE over 1979–2020, due to the superior quality of global re-analyses onward from 1979. The SST timeseries used in the quantile regressions is ERSSTv5[22] SST fields, averaged over 10–36 °N, 100–70 °W excluding the portions of this box over the Pacific Ocean. The spatial resolution of ERSSTv5 is 2°. RH, 200 T, and VWS timeseries are calculated from April and May monthly mean ERA5[21] fields. The primary GPI timeseries used in this study[23] is calculated from April and May monthly mean ERA5 fields. The GPI value is calculated at each 0.25° gridpoint in this box per the equation[23,25]:

$$GPI = abs(10^5\eta)^2 * \left(\frac{RH}{50}\right)^3 * \left(\frac{PI}{70}\right)^3 * (1 + 0.1VWS)^{-2} \quad (3)$$

with η the absolute 850-hPa vorticity, RH the 700-hPa RH, PI the potential intensity, a theoretical TC intensity maximum given SST and a specified

atmospheric column[25,45], and VWS is the vertical wind shear between 850 and 250 hPa. The GPI statistic was developed using a fitting procedure of these variables in the NCEP/NCAR Reanalysis[46] to the seasonal and spatial climatological record of global cyclogenesis events. These gridpoint GPI values are then averaged over 10–36°N, 100–70°W separately for April and May, and the two months are then averaged to yield the GPI timeseries used in the quantile regressions.

An alternative form of GPI[24] was used at several points in this study for comparative purposes using the same data sources described above. This construction of GPI is of the form:

$$GPI = |\eta|^3 * \chi^{-\frac{4}{3}} * (MAX((PI - 35ms^{-1}), 0)^2 * (25ms^{-1} + VWS)^{-4} \quad (4)$$

with η, PI, and VWS as above, and

$$\chi = \frac{s_b - s_m}{s_0^* - s_b} \quad (5)$$

with χ a measure of moist entropy deficit in the middle troposphere and $s_b$, $s_m$, and $s_0^*$ moist entropies of the boundary layer and middle troposphere, and saturation moist entropy of the sea surface[24].

*TC genesis SST threshold and SST trends.* Because daily SST values are not available from the ERSSTv5 dataset, spatial coverage of the 26.5 °C threshold value is calculated from daily mean ERA5[21] SST fields, which have a spatial resolution of 0.25°. ERA5 SSTs are drawn from several different analyses over the period 1950-2020, including HadISST2.1.0.0 prior to 1961, HadISST2.1.1.0 between 1961 and mid-2007[47], and OSTIA since mid-2007[48]. The native temporal resolutions of these analyses are one month, five days, and one day, respectively. However, because SSTs change slowly with time, using these datasets downscaled to a uniform daily resolution is unlikely to introduce significant observational bias into the SST threshold analysis. The proportion of the 10–36°N, 100–70°W box for which this criterion is satisfied is calculated by applying a land mask and excluding the Pacific Ocean and dividing the number of daily gridpoints exceeding 26.5 °C by the total number of non-land, non-Pacific gridpoints. Trends of ERSSTv5 monthly mean

SSTs are calculated from ordinary least squares regressions against year. Box or global average SST trends are calculated after excluding land gridpoints. Supplementary Fig. 7 was the only instance in which ERA5 SST values rather than ERSSTv5 monthly means were used as a source of SST values.

*Optimized hurricane season bounds.* Objective start and end dates for Atlantic hurricane season are calculated from 50-year trailing average timeseries of named storm formations, CONUS named storm impacts, Atlantic named storm days, Atlantic ACE, and USACE, with a 15-day intraseasonal smoothing filter applied, normalized by total activity within the window. This results in a daily timeseries of the smoothed percentage of total TC activity occurring each day from 1 March–28/29 February within the 50-year window. A middle-out algorithm begins at the climatological peak of Atlantic TC activity in mid-September and adds one day at a time to the objective season until the period encompassing the fewest contiguous days for a specified percentage activity threshold is met. The thresholds tested are 95%, 97%, and 99% of total activity for each of the TC metrics.

## Data availability

Cited source data is available at the following; National Hurricane Center Tropical Cyclone Reports: https://www.nhc.noaa.gov/data/#tcr. ERSSTv5: https://www.ncei.noaa.gov/pub/data/cmb/ersst/v5/; HURDAT2: https://www.aoml.noaa.gov/hrd/hurdat/hurdat2.html; ERA5: https://cds.climate.copernicus.eu/#!/search?text=ERA5&type=dataset.

## Code availability

All scripts and data used to generate the figures shown in this paper are freely available at http://www.weathertiger.com/seasonshiftcode. These scripts may not be used for commercial purposes.

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

## Acknowledgements

This work has been supported by the G. Unger Vetlesen Foundation (P.J.K.), National Science Foundation award AGS-2011812 (K.M.W.), NOAA awards NA19OAR4590135 and NA18NWS4680066 (D.J.H.), and NOAA via the Cooperative Institute for Satellite Earth System Studies under Cooperative Agreement NA19NES4320002 (C.J.S.).

## Author contributions

R.E.T., P.J.K., E.M.S., and C.J.S. conceptualized and designed the study. R.E.T., P.J.K., and K.M.W. curated the data and performed formal analysis. R.E.T. performed the investigation and created the visualizations using a methodology developed by R.E.T., P.J.K., E.M.S., K.M.W., and D.J.H. E.M.S. and R.E.T. wrote the original draft, with R.E.T., P.J.K., E.M.S., K.M.W., D.J.H, C.J.S., and E.S.B. participating in the review and editing process.

## Competing interests

The authors declare no competing interests.
