## [Peer Review File · Nature Communications]

Earlier onset of North Atlantic hurricane season with warming oceansREVIEWER COMMENTS

Reviewer #1 (Remarks to the Author):

Please see the attached review. I have enclosed an unformatted text version below.

**Review of NCOMMS-21-45142-T by Truchelut et al.
Clark Evans, 11 December 2021**

Overview: This is a well-written examination of long-term trends in metrics related to Atlantic hurricane season activity. The authors find that (1) the first tropical cyclone in each season is forming earlier in the year, (2) this trend also manifests in other related metrics of tropical cyclone activity, and (3) this trend is predominantly driven by warming sea-surface temperatures in the tropical cyclones' climatologically favored formation regions.

As my concerns with the manuscript are relatively minor, I recommend publication after the authors have had the opportunity to address a few minor points.

Overarching Comments

1. I appreciate that the manuscript evaluates trends in several metrics, at several thresholds, that could be used precisely define the hurricane season. These metrics can broadly be classified as occurrence-based (as are used to define most meteorological climatologies) or impacts-based (emphasizing societal impacts). I recommend making these classifications more explicit within the narrative, particularly since the long-term trend magnitudes vary somewhat between the occurrence- and impacts-based metrics considered.

Furthermore, the impact-based metrics do not explicitly consider TC rainfall, which is often the most-significant impact from early- and late-season TCs (as implied by Line 25), and it remains an open question in the social sciences as to whether and how different publics conceptualize hazard seasonality. Consequently, the considered metrics do not fully capture the problem's physical and societal dimensions. I recommend adding 1-2 sentences to the Discussion section to acknowledge these limitations and thus motivate further study.

2. The most-significant trends in the metrics and parameters considered are found in the first- through third-percentile dates of each considered metric. The first-percentile date is uniquely attributable to the date of the season's first TC formation, while the second- and third-percentile dates are significantly influenced by the date of the season's first TC formation. In other words, these results appear to be influenced primarily by changes in the season's first TC formation event rather than a more-extensive shift in the season's earliest-forming TCs. I am not sure that this comes across as strongly in the manuscript as it could, however, and thus I recommend revising the text (particularly in the discussion section) to better address this point.

Specific Comments

1. Lines 48-50: This sentence is confusing. Are you trying to say that 17% of all USACE occurs before August 1st, with only 6% of total ACE occurring before August 1st? Or that only 6% of total ACE occurs near or over the United States? I think it's the former, but revising this sentence would be appreciated to alleviate the confusion.

2. Lines 107-110: It is peculiar that the area indicated in Supplementary Figure 4 excludes the high-density region of early-season TC formations immediately off the southeast United States coastline. I recommend revising the analyses that are based on

the area indicated in Supplementary Figure 4 to be based on a larger area extending to 35°N (or 34°N/36°N given the ERSSTv5's 2° horizontal grid spacing). I expect that expanding this area will further support the associated analyses given the large positive trend in sea-surface temperatures over this region, but explicit quantification is needed to confirm this expectation.

3. Lines 119-130: This paragraph appears to have two emphases: quantifying how ACE percentile dates vary as a function of April-May GPI (e.g., interannual variability) and quantifying trends in ACE percentile dates as a function of April-May GPI trends (e.g., long-term variability). I found this paragraph somewhat difficult to digest because of its attempt to describe these distinct types of variability. To ameliorate this concern, I recommend splitting the text into two paragraphs, one covering each type of variability, similar to how the two paragraphs currently following this one (interannual variability in GPI parameters in the first, long-term variability related to GPI parameter trends in the second) are currently structured.

4. Line 239: The correct title for this article is "Counting Atlantic Tropical Cyclones Back to 1900" (available at <https://agupubs.onlinelibrary.wiley.com/doi/epdf/10.1029/2007EO180001>).

5. Supplemental Material, Line 12: I believe this should be "As in Fig. 1a and 1b" rather than "As in Fig. 1b and 1c."

6. Supplemental Material, Lines 74, 83, 89, and 96: I believe that spaces should be added between the time and UTC for both 0000UTC and 2359UTC in the lines where each are found.

7. Supplemental Material, Line 121: It may be a rendering issue with the PDF on my Mac, but the eta that appears in the GPI equation is replaced by a square box in this line.

8. Supplemental Material, Lines 129-137: The ERA5 SST analysis is drawn from different sources as a function of time, as given by Table 7 of Hersbach et al. (2020). Although ERA5 SST data are provided to end-users at a daily frequency, data from prior to 1961 are based on monthly analyses and data from 1961 to mid-2007 are based on pentad analyses. I recommend revising the supplemental material to briefly acknowledge both the evolving analysis frequency and changeovers in SST analysis (HadISST2.1.0.0 -> HadISST2.1.1.0 -> OSTIA) that occur in ERA5 from 1950 onward.

Reviewer #2 (Remarks to the Author):

The manuscript "Earlier onset of North Atlantic hurricane season with warming oceans" explores the earlier onset of North Atlantic hurricane season found in observations and argues that it is caused by warming sea surface temperature, especially over the western Atlantic basin (WATL). It is both scientifically interesting and practically important for hurricane-impacted regions. However, the analyses that link the earlier onset to pre-season environment change are not convincing and need to be improved.

Major comments:

1. The GPI used in the study might not be the proper TC genesis index to use. Emanuel (2010) argues that it is entropy deficit, not relative humidity, that is relevant to TC genesis and therefore proposes the new version of GPI. Furthermore, recent studies (e.g. Wang and Murakami 2020) emphasizes the importance of dynamical factors, especially the vertical motion, in controlling TC genesis. The behind mechanism can be understood under the framework of seeds and genesis probability (Tsieh et al. 2020, Yang et al. 2021). It would be insightful to look into these new TC genesis indices or controlling factors.

2. Besides the choice of TC genesis indices or controlling factors, the relationship between TC season onset and GPI used in this study is still not clear from the linear regression. Given both variables have significant linear trend during the same period, it is not surprising they correlate well with each other. The question is whether the onset date and GPI are still correlated after both being de-trended, which could serve as a test on the argument the onset date is controlled by GPI variability.

3. To look at contributions to the GPI trend from individual components, the authors seem to apply a linear regression or multi-linear regression framework, which is additive in nature. However, these individual components are combined in a multiplicative way to form GPI. The authors need to justify why the linear additive way works.

References:

**Emanuel, K (2010): Tropical cyclone activity downscaled from NOAA-CIRES reanalysis, 1908–1958. JAMES
<https://agupubs.onlinelibrary.wiley.com/doi/full/10.3894/JAMES.2010.2.1>.**

Hsieh TL et al. Large-scale control on the frequency of tropical cyclones and seeds: a consistent relationship across a hierarchy of global atmospheric models. Clim Dyn 55, 3177–3196 (2020). <https://doi.org/10.1007/s00382-020-05446-5>.

Wang B and H Murakami (2020): Dynamic genesis potential index for diagnosing present-day and future global tropical cyclone genesis. Environ. Res. Lett. 15, 114008.

Yang W, TL Hsieh, GA Vecchi (2021): Hurricane annual cycle controlled by both seeds and genesis probability. Proceedings of the National Academy of Sciences. 118 (41) e2108397118.

Author Response to reviewers of “Earlier onset of North Atlantic hurricane season with warming oceans”

General Response: We appreciate the thoughtful feedback and suggestions proposed by both reviewers, and thank them for their conditional support for publication in Nature Communications. We have undertaken significant revisions of the manuscript to address these comments and concerns, and we believe that the suggested revisions have made the manuscript stronger and clearer. If there are remaining questions concerning the revised manuscript, we look forward to addressing them. Detailed and specific responses to the reviewers’ comments are provided below, following each comment.

Review of NCOMMS-21-45142-T by Truchelut et al.

Clark Evans, 11 December 2021

Overview: This is a well-written examination of long-term trends in metrics related to Atlantic hurricane season activity. The authors find that (1) the first tropical cyclone in each season is forming earlier in the year, (2) this trend also manifests in other related metrics of tropical cyclone activity, and (3) this trend is predominantly driven by warming sea-surface temperatures in the tropical cyclones’ climatologically favored formation regions.

As my concerns with the manuscript are relatively minor, I recommend publication after the authors have had the opportunity to address a few minor points.

We are very appreciative of the insight and thoughtfulness of the reviewer’s comments and suggestions, and will attempt to address each of them fully here.

Overarching Comments

1. I appreciate that the manuscript evaluates trends in several metrics, at several thresholds, that could be used precisely define the hurricane season. These metrics can broadly be classified as occurrence-based (as are used to define most meteorological climatologies) or impacts-based (emphasizing societal impacts). I recommend making these classifications more explicit within the narrative, particularly since the long-term trend magnitudes vary somewhat between the occurrence- and impacts-based metrics considered.

Furthermore, the impact-based metrics do not explicitly consider TC rainfall, which is often the most-significant impact from early- and late-season TCs (as implied by Line 25), and it remains an open question in the social sciences as to whether and how different publics conceptualize hazard seasonality. Consequently, the considered metrics do not fully capture the problem’s physical and societal dimensions. I recommend adding 1-2 sentences to the Discussion section to acknowledge these limitations and thus motivate further study.

We agree that all of the metrics considered in this study are, at best, rough proxies for the societal impact of TCs, and it is this reason that motivated us to consider a wide range of measures of TC risk onset. To clarify our approaches, the occurrence- and impacts-based dialectic has been adopted where appropriate throughout the revised manuscript. The discussion section now highlights the limitations of these metrics as imperfect proxies for societal impacts of TCs, and qualitatively notes

several massive U.S. flooding events that have historically been caused by TCs forming early in the season.

2. The most-significant trends in the metrics and parameters considered are found in the first- through third-percentile dates of each considered metric. The first-percentile date is uniquely attributable to the date of the season's first TC formation, while the second- and third-percentile dates are significantly influenced by the date of the season's first TC formation. In other words, these results appear to be influenced primarily by changes in the season's first TC formation event rather than a more-extensive shift in the season's earliest-forming TCs. I am not sure that this comes across as strongly in the manuscript as it could, however, and thus I recommend revising the text (particularly in the discussion section) to better address this point.

While using the ACE-based percentile metrics offers some methodological protection against confounding cyclical (or secular) trends in overall activity, we agree that the primary changes in climatology over 1979-2020 appear to be driven primarily by trends in the initial formation dates of the first one or two TCs per year (as well as initial CONUS landfall dates). The text has been updated at several points to clarify that this is the source of the climatological changes, including in the "season onset trend assessment" section relating the quantile regressions to the values of the trendline slopes from Supplemental Figure 1, as well as reiterating this point in the first and last paragraph of the discussion.

Specific Comments

1. Lines 48-50: This sentence is confusing. Are you trying to say that 17% of all USACE occurs before August 1st, with only 6% of total ACE occurring before August 1st? Or that only 6% of total ACE occurs near or over the United States? I think it's the former, but revising this sentence would be appreciated to alleviate the confusion.

This sentence has been rewritten to clarify that it is the former: over 1979-2020, 6% of basinwide ACE occurs prior to 1 August, but 17% of ACE near the CONUS (USACE) occurs prior to 1 August.

2. Lines 107-110: It is peculiar that the area indicated in Supplementary Figure 4 excludes the high-density region of early-season TC formations immediately off the southeast United States coastline. I recommend revising the analyses that are based on the area indicated in Supplementary Figure 4 to be based on a larger area extending to 35°N (or 34°N/36°N given the ERSSTv5's 2° horizontal grid spacing). I expect that expanding this area will further support the associated analyses given the large positive trend in sea-surface temperatures over this region, but explicit quantification is needed to confirm this expectation.

We agree that the larger box is more representative of the geographic extent of pre-season and early season TC genesis, particularly within the past decade. The region over which the GPI and all thermodynamic and dynamic synoptic variables are averaged has been expanded to 10-36°N, 100-70°W for all analyses in this version of the manuscript. This change has resulted in slightly different values for trends and significances in the "quantifying environmental changes" and discussion sections of the paper, but there are no major differences in the key results of the paper. Overall, including the

additional portion of the western Atlantic subtropics slightly strengthened the relationships between higher RH/SST/GPI and earlier season onset found in the previous draft of the manuscript.

3. Lines 119-130: This paragraph appears to have two emphases: quantifying how ACE percentile dates vary as a function of April-May GPI (e.g., interannual variability) and quantifying trends in ACE percentile dates as a function of April-May GPI trends (e.g., long-term variability). I found this paragraph somewhat difficult to digest because of its attempt to describe these distinct types of variability. To ameliorate this concern, I recommend splitting the text into two paragraphs, one covering each type of variability, similar to how the two paragraphs currently following this one (interannual variability in GPI parameters in the first, long-term variability related to GPI parameter trends in the second) are currently structured.

We concur that this change makes the analysis clearer, and have separated the content into two paragraphs. We have also added a review of the long-term variability for USACE into the second paragraph.

4. Line 239: The correct title for this article is "Counting Atlantic Tropical Cyclones Back to 1900" (available at <https://agupubs.onlinelibrary.wiley.com/doi/epdf/10.1029/2007EO180001>).

We have corrected the title of the article.

5. Supplemental Material, Line 12: I believe this should be "As in Fig. 1a and 1b" rather than "As in Fig. 1b and 1c."

Thank you for catching this. We have changed the sentence accordingly.

6. Supplemental Material, Lines 74, 83, 89, and 96: I believe that spaces should be added between the time and UTC for both 0000UTC and 2359UTC in the lines where each are found.

We have now included spaces per your suggestion.

7. Supplemental Material, Line 121: It may be a rendering issue with the PDF on my Mac, but the eta that appears in the GPI equation is replaced by a square box in this line.

We have ensured that the eta appears on both Mac and Windows machines in the corrected render.

8. Supplemental Material, Lines 129-137: The ERA5 SST analysis is drawn from different sources as a function of time, as given by Table 7 of Hersbach et al. (2020). Although ERA5 SST data are provided to end-users at a daily frequency, data from prior to 1961 are based on monthly analyses and data from 1961 to mid-2007 are based on pentad analyses. I recommend revising the supplemental material to briefly acknowledge both the evolving analysis frequency and changeovers in SST analysis (HadISST2.1.0.0 -> HadISST2.1.1.0 -> OSTIA) that occur in ERA5 from 1950 onward.

Several sentences and two citations have been added to the supplemental material detailing the provenance of the SST analyses used by ERA5 and the various time-averaging windows of those analyses, as well as justifying our choice of the dataset for the SST threshold study.

Once again, thank you for your comments and suggestions, and for your contributions to improving our paper.

Reviewer #2 (Remarks to the Author):

The manuscript “Earlier onset of North Atlantic hurricane season with warming oceans” explores the earlier onset of North Atlantic hurricane season found in observations and argues that it is caused by warming sea surface temperature, especially over the western Atlantic basin (WATL). It is both scientifically interesting and practically important for hurricane-impacted regions. However, the analyses that link the earlier onset to pre-season environment change are not convincing and need to be improved.

Thank you for your insightful comments and feedback on our manuscript. We have taken your suggestions and in tandem with the suggestions of reviewer #1, we believe that the major revisions undertaken have significantly improved the quality of the paper. We hope that these changes will satisfactorily address your concerns.

Major comments:

1. The GPI used in the study might not be the proper TC genesis index to use. Emanuel (2010) argues that it is entropy deficit, not relative humidity, that is relevant to TC genesis and therefore proposes the new version of GPI. Furthermore, recent studies (e.g. Wang and Murakami 2020) emphasizes the importance of dynamical factors, especially the vertical motion, in controlling TC genesis. The behind mechanism can be understood under the framework of seeds and genesis probability (Tsieh et al. 2020, Yang et al. 2021). It would be insightful to look into these new TC genesis indices or controlling factors.

We appreciate the comment here and the opportunity to connect our analysis more directly to a clear physical interpretation of the results. Based on this request, we have repeated analyses involving the Camargo (2007) GPI in the “Quantifying environmental changes” section with the suggested Emanuel (2010) GPI. Repeating the tests shown in Figure 2 with the Emanuel GPI, we found similar findings to using the C07 GPI for correlation coefficients between USACE and ACE initial percentile threshold dates. The overall implied change in season onset using E10 GPI is less than that calculated using C07 GPI, as there is less of a trend higher in E10 GPI values over the period, though all relationships retain their same directionality. The E10 GPI parameter trends over 1979-2020 and regression coefficients with first percentile ACE and USACE threshold dates over the same period have been added to Supplementary Table 1 for comparison. We have also noted at several places in the text that the two formulations of GPIs yielded similar results, and added descriptive material for E10 GPI to the supplementary method section, including a discussion of the inclusion of entropy deficit rather than relative humidity. Thank you for this suggestion, as inclusion of multiple GPIs will increase confidence that the results are not simply an artifact of the specific index utilized in the study.

We appreciate the importance of the new research into seeding as a control on cyclogenesis frequency and believe this would be a rewarding direction for a future study of intraseasonal distribution of activity. However, in the case of season onset, AEWs are not usually the incipient disturbances triggering TC development in May and June, so there was not a clear connection between those valuable studies and this work.

2. Besides the choice of TC genesis indices or controlling factors, the relationship between TC season onset and GPI used in this study is still not clear from the linear regression. Given both variables have

significant linear trend during the same period, it is not surprising they correlate well with each other. The question is whether the onset date and GPI are still correlated after both being de-trended, which could serve as a test on the argument the onset date is controlled by GPI variability.

This point is well-taken, and the revised version of the paper now explicitly includes results for the relationship between detrended onset date of initial ACE percentiles over 1979-2020 (both with and without including short-lived TCs in the calculation) and detrended spring Western Atlantic SSTs. Because GPI is a highly-derived parameter, we felt that the detrended April-May SSTs was more representative of the link between anomalous season onset and anomalous thermodynamical environmental conditions. The significance of the relationship between initial percentiles of ACE with and without short-lived TCs is $p < 0.05$ over 1979-2020, in the expected direction of higher SSTs leading to earlier season onset. These results are now reported in the revised manuscript as supporting evidence in the discussion section. The p-value between mean detrended WATL May GPI and the initial three detrended ACE percentile threshold dates lies outside statistical significance ($p = 0.15$), but the complex derivation of GPI and the relatively short timeframe (~40 years) of the trustworthy dataset likely mean that additional observations are needed for this relationship to become clearer. In this case, we believe that the SST trend itself is also physically significant in terms of modifying the net thermodynamical environment, which we argue occurs via the physical mechanism of increased column instability in the discussion section.

3. To look at contributions to the GPI trend from individual components, the authors seem to apply a linear regression or multi-linear regression framework, which is additive in nature. However, these individual components are combined in a multiplicative way to form GPI. The authors need to justify why the linear additive way works.

Thank you for making this point, as we did not mean to imply that the individual dynamic and thermodynamic parameters included in the GP indices as well as other genesis frameworks such as Demaria et al 2001 were linearly additive. As noted in Camargo et al 2007: "Owing to the nonlinearity of the GP index, the net anomaly cannot be described as the sum of the four fields described here. Nonetheless, to the extent that the index provides weights that appropriately quantify the roles of the different factors in genesis and the nonlinearities are not too large, the attributions obtained by this method should be meaningful." Therefore, we have removed all references to the concept of the contributions of the individual GPI components being cumulative. A greater emphasis has been placed on the estimated season onset shift calculated from the SST regression onto the observed changes in initial TC development and CONUS landfall activity, in line with the methodology in previous papers as cited above.

Thanks again! We appreciate your thoughtful feedback and comments on our paper.

References:

Emanuel, K (2010: Tropical cyclone activity downscaled from NOAA-CIRES reanalysis, 1908–1958. JAMES <https://agupubs.onlinelibrary.wiley.com/doi/full/10.3894/JAMES.2010.2.1>.

Hsieh TL et al. Large-scale control on the frequency of tropical cyclones and seeds: a consistent relationship across a hierarchy of global atmospheric models. *Clim Dyn* 55, 3177–3196 (2020). <https://doi.org/10.1007/s00382-020-05446-5>.

Wang B and H Murakami (2020): Dynamic genesis potential index for diagnosing present-day and future global tropical cyclone genesis. *Environ. Res. Lett.* 15, 114008.

Yang W, TL Hsieh, GA Vecchi (2021): Hurricane annual cycle controlled by both seeds and genesis probability. *Proceedings of the National Academy of Sciences.* 118 (41) e2108397118.

REVIEWERS' COMMENTS

Reviewer #2 (Remarks to the Author):

The authors have addressed the concerns from the reviewer.